# Cardiac Electromechanical Activity in Healthy Cats and Cats with Cardiomyopathies

**DOI:** 10.3390/s23198336

**Published:** 2023-10-09

**Authors:** Maja Brložnik, Ema Lunka, Viktor Avbelj, Alenka Nemec Svete, Aleksandra Domanjko Petrič

**Affiliations:** 1Faculty of Health Sciences, University of Ljubljana, 1000 Ljubljana, Slovenia; maja.brloznik@zf.uni-lj.si; 2Small Animal Clinic, Veterinary Faculty, University of Ljubljana, 1000 Ljubljana, Slovenia; el9314@student.uni-lj.si (E.L.); alenka.nemecsvete@vf.uni-lj.si (A.N.S.); 3Department of Communication Systems, Jožef Stefan Institute, 1000 Ljubljana, Slovenia; viktor.avbelj@ijs.si

**Keywords:** feline, electrocardiography (ECG) synchronized to phonocardiography (PCG), echocardiography, cardiomyopathy

## Abstract

Optimal heart function depends on perfect synchronization between electrical and mechanical activity. In this pilot study, we aimed to investigate the electromechanical activity of the heart in healthy cats and cats with cardiomyopathy with phonocardiography (PCG) synchronized to an electrocardiography (ECG) pilot device. We included 29 cats (12 healthy cats and 17 cats diagnosed with cardiomyopathy) and performed a clinical examination, PCG synchronized with ECG and echocardiography. We measured the following durations with the pilot PCG device synchronized with ECG: QRS (ventricular depolarization), QT interval (electrical systole), QS1 interval (electromechanical activation time (EMAT)), S1S2 (mechanical systole), QS2 interval (electrical and mechanical systole) and electromechanical window (end of T wave to the beginning of S2). The measured parameters did not differ between healthy cats and cats with cardiomyopathy; however, in cats with cardiomyopathy, EMAT/RR, QS2/RR and S1S2/RR were significantly longer than in healthy cats. This suggests that the hypertrophied myocardium takes longer to generate sufficient pressure to close the mitral valve and that electrical systole, i.e., depolarization and repolarization, and mechanical systoles are longer in cats with cardiomyopathy. The PCG synchronized with the ECG pilot device proved to be a valuable tool for evaluating the electromechanical activity of the feline heart.

## 1. Introduction

Auscultation of the heart provides rapid diagnostic information of heart sounds and rhythm, while using phonocardiography (PCG), we record heart sounds and provide graphic information for further analysis. The temporal relationship between mechanical and electrical events can be determined by accurate simultaneous recording and measurement with PCG and electrocardiography (ECG). The analysis of digitally recorded heart sounds and simultaneous ECG was first investigated in 2003 [1]. Synchronization of PCG with ECG is important because it allows accurate analysis of the various electromechanical relations. Electromechanical activation time (EMAT or QS1 interval) is the time from the onset of electrical activity (Q wave) in the ventricles until the closure of the mitral and tricuspid valves or first heart sound (S1). In human patients with congestive heart failure (CHF), EMAT is used to predict adverse cardiac events and discharge outcomes [2,3,4] and has been shown to be strongly associated with impaired left ventricular contractility but not with LV filling pressure [3,5]. 

The interval QT (from the onset of the Q wave to the end of the T wave) is the electrical systole, while the interval between S1 and the closure of the aortic and pulmonic valves or the second heart sound (S2), i.e., S1S2, is the mechanical systole. Although ventricular electrical activity always precedes mechanical activity, the end of mechanical activity may precede or follow the end of the T wave, indicating the end of the electrical systole. The electromechanical window (EMW) is the time interval between the end of the T wave and the closure of the aortic and pulmonic valves and has a positive value when the T wave ends before valve closure and vice versa. A negative EMW would mean that ventricular relaxation occurs earlier than electrical repolarization and is most commonly associated with long QT syndrome in humans. The electromechanical window (EMW) characterizes the synchrony between ventricular mechanical activity and electrical repolarization. The temporal shift of mechanical activity relative to electrical events can lead to arrhythmias [6,7,8].

In veterinary medicine, there are little scientific data on electromechanical durations, apart from the results of invasive measurements in experimentally induced long QT in dogs [7] and recordings in healthy horses [9] and dogs with different stages of myxomatous mitral valve disease [10]. Feline cardiomyopathies are the most prevalent cardiac disease in adult cats, and feline hypertrophic cardiomyopathy (HCM) is the most common among them [11,12,13]. In cats with left ventricular hypertrophy, ECG analysis may have predictive and prognostic value [14].

Heart rate affects the durations of electromechanical durations, and therefore they are normalized to the cardiac cycle. Generally, the RR interval is used to correct EMAT to heart rate. EMAT/RR is a fraction of EMAT in the cardiac cycle and reflects the time required for the left ventricular systole to generate sufficient pressure to close the mitral valve [2,3,4,5]. A QRS-RR regression slope is used to correct for QRS [15,16]. In addition, the QT interval is corrected by several formulas using a logarithmic, hyperbolic or exponential function, but they all have their limitations [17,18]. The Bazett formula, QT/square root of RR [19] was the first formula for QTc, and although several errors of the Bazett formula have been identified, and the Fridericia correction, QT/cubic root of RR [20], is used in clinical trials of new drugs [21], studies still support its use for infants and children [17]. All these formulas do not take into account the influence of autonomic tone on the QT interval, independent of the effects of heart rate, and the relatively slow adaptation of repolarization to changes in heart rate. In veterinary medicine, the inverse relationship between the QT interval and the RR interval was investigated in a 24 h ECG study in 14 healthy beagle dogs [22] and 20 healthy cats [23]. In the study with 14 beagle dogs, the authors proposed a logarithmic formula to correct QT for heart rate, QTc = log600 × QT/logRR [22], which was later applied in a study with 22 healthy cats and 35 cats with left ventricular hypertrophy [14]. In the study with 20 healthy cats, the authors proposed a formula predicting QT based on heart rate: QTpredicted = 0.41845798 − (0.00181963 × HR) + (0.00000313 × HR^2^) [23].

We aimed to evaluate cardiac electromechanical activity in healthy cats and cats with cardiomyopathies using a digital high-resolution ECG device with wireless data transmission synchronized with a non-commercial PCG pilot device. We looked for possible differences in electromechanical activity between cats with cardiomyopathy and healthy cats in order to detect possible coupling abnormalities between electrical and mechanical events in the feline heart.

## 2. Materials and Methods

### 2.1. Animals

In a prospective pilot study, we examined 45 cats that were presented for vaccination and/or health check or admitted as patients. We performed the following procedures: clinical examination, ECG synchronized with PCG, and echocardiography.

### 2.2. Electrocardiography

For ECG recordings (with a sampling rate of 500 Hz and a resolution of 0.05 μV), we used a digital high-resolution 12-lead ECG device with wireless data transmission (Cardiax PC-ECG, MESA Medizintechnik GmbH, Benediktbeuern, Germany) connected to a laptop (Windows 11) via a battery-powered ECG sensor WiFi. A Digital Diagnostic Center (MESA Medizintechnik GmbH, Benediktbeuern, Germany, v 2.4.0) ran on the computer during recording for visual inspection and export of data to HL7 format for further processing. A six- to nine-lead ECG was recorded simultaneously.

### 2.3. Phonocardiography

To record heart sounds, we used a PCG device with a microphone inserted into the acoustic stethoscope tube and connected to a smartphone (Huawei P30 lite, Huawei, Shenzhen, China). The head of the stethoscope was positioned on the left side of the chest, where the S1 and S2 heart sounds were best heard. To ensure good recording quality, the head of the stethoscope was then held as still as possible for at least 10 s. Sound recordings were conducted using the WaveEditor application for Android (Atlanta, GA, USA, v 1.82) with a sampling rate of 48 kHz.

### 2.4. Synchronization of Electrocardiographic and Phonocardiographic Data

Given that the PCG and ECG data ran on separate free-running oscillators, it was necessary to synchronize the recordings [24,25,26]. To achieve this, we built a synchronization device that generates an acoustic test signal for the PCG recording and an electrical one for the ECG. For synchronization purposes, the stethoscope head was 1 cm away from the sound source and ECG electrodes were attached to the synchronization device. We recorded the test signals at the beginning and end of each measurement. This allowed us to accurately synchronize the PCG and ECG recordings with linear time interpolation.

The first step of the analysis was accurate time synchronization of the recordings obtained from the independent devices (ECG, PCG). The synchronization device was used to generate three consecutive signals at the beginning of the recording, before auscultation of the heart, and three consecutive signals after auscultation of the heart. For time synchronization in the analysis, we selected the third signal at the beginning of the recording and the third at the end of the recording to the nearest millisecond. Aligning the beginning and end of the synchronization pulse and linear time interpolation enabled an accurate synchronization of the recordings.

The next step was to determine the 10 s intervals of the recordings with the best quality ECG and PCG signals. From the selected intervals, the following time points were determined for each heartbeat: from the ECG recording, the onset of the QRS, peak of the R wave and end of the T wave; and from the PCG recording, the onset of the S1 tone and the onset of the S2 tone. From these time points, the following intervals were calculated: QRS, QT, EMAT, QS2, EMW, RR and S1S2 for six consecutive heartbeats. 

Measurements of electrical events were performed using the VisECG program (Jožef Stefan Institute, Ljubljana, Slovenia) and the Digital Diagnostic Center. For the analysis with the VisECG program, we exported the recordings from the DDC Digital Diagnostic Center program in HL7 format to allow analysis with the VisECG program. For the beginning of QRS, an unfiltered version of the ECG was used. We considered the beginning of the QRS at the Q or R wave start (in the ECG recordings of cats, the Q wave was mostly not visible on the recorded leads). The end of the T wave was defined using the tangent method at the intersection of the downward tangent with the baseline. Heart rate was calculated from the five RR intervals used for PCG and synchronized with ECG measurements. Tachycardia was diagnosed if the heart rate exceeded 240 beats per minute [27]. PCGs were analyzed using open-source, cross-platform audio software Audacity program (Muse Group, Limassol, Cyprus, v 2.2.2).

We normalized the QT interval to the cardiac cycle by various formulas for QTc [14,19,20,22]:QTc1 = log600 × QT/logRR 
QTc2 = log300 × QT/logRR
QTc3 = QT/cube root of RR 
QTc4 = QT/square root of RR

Additionally, we calculated QT predicted from heart rate [23] and the difference between the measured and predicted QT.
QTpredicted = 0.41845798 − (0.00181963 × HR) + (0.00000313 × HR^2^)

For all electromechanical and mechanical durations, we calculated their share in the cardiac cycle by dividing them by the RR interval, as was performed for EMAT in previous studies [2,3,4,5].

### 2.5. Echocardiography

A complete echocardiographic examination of the heart was performed by two experienced veterinarians (ADP, MB) according to the guidelines [28]. A Vivid E9 echocardiographic system (General Electrics Healthcare, Milwaukee, WI, United States) with a 5–10 MHz transducer was used. A limb ECG lead II was recorded simultaneously. Loops and images were analyzed using an offline Echo-PAC workstation (GE Healthcare, Europe). In all cats, the following echocardiographic parameters were measured: two-dimensional left atrial measurements from the right parasternal long and short-axis view; left ventricular M-mode or 2-D measurements from the right parasternal short-axis view; spectral and color flow Doppler velocities from the left apical view; and tissue Doppler annular velocities from the left apical view.

Normal cats were considered according to previously published feline echocardiographic references, and hypertrophic cardiomyopathy was diagnosed if the LV wall was thicker than 6 mm [29,30,31]. All animals with systemic hypertension, hyperthyroidism, aortic stenosis, neoplastic infiltration or other congenital and acquired diseases were excluded from the study. Restrictive cardiomyopathy was diagnosed when normal LV walls with biatrial enlargement and restrictive mitral inflow pattern was detected [32,33].

### 2.6. Statistical Analysis

Statistical analysis was performed with a commercially available computer program (IBM SPSS 28, Chicago, IL, USA). The Shapiro–Wilk test was performed to determine the distribution of the data. Normally distributed data, expressed as mean and standard deviation (SD), were compared between healthy cats and cats with cardiomyopathy using the independent t-test. Non-normally distributed data, expressed as median and interquartile range (IQR, 25–75th percentile), were compared using the Mann–Whitney test. Statistical significance was set at *p* < 0.05.

## 3. Results

We initially examined 45 cats, but later excluded 16 of them (11 were diagnosed with other cardiac diseases, and five cats were sedated). Ultimately, 29 cats were included in the study, and their characteristics are shown in Table 1. Echocardiographic analysis revealed that 12 cats had normal heart size and function, and 17 cats were diagnosed with cardiomyopathy (Table 1). Among the cats with cardiomyopathy, four had CHF (23.5%, 4/17) and 13 did not have CHF (five with enlarged left atrium (29.4%, 5/17) and eight with normal dimensions of the cardiac chambers (47.1%, 8/17).

Age and weight were not significantly different between healthy cats and cats with cardiomyopathy (*p* = 0.063 and *p* = 0.659, respectively).

Of the 17 cats diagnosed with cardiomyopathy, 14 were diagnosed with HCM and three had restrictive cardiomyopathy.

Atrial fibrillation was diagnosed in two cats with cardiomyopathy (11.8%, 2/17), and sinus tachycardia was diagnosed in two cats with cardiomyopathy (11.8%, 2/17). In both cats with sinus tachycardia, premature ventricular and fusion complexes were present. Sinus rhythm was present in 13 cats with cardiomyopathy (76.5%, 13/17). One cat in sinus rhythm had one episode of paroxysmal supraventricular tachycardia. In the group of healthy cats, sinus rhythm was diagnosed in 12 cats (100%, 12/12).

Phonocardiographic data synchronized with ECG are shown in Table 2. Three consecutive signals at the beginning of the recording, before auscultation of the heart, were used for synchronization of the recordings, as shown in Figure 1. The unfiltered and filtered versions of the lead III in the ECG are shown in Figure 2. Representative images of synchronized ECG and PCG recordings in a healthy cat and a cat with cardiomyopathy are shown in Figure 3 and Figure 4, respectively.

In seven cats (24.1%, 7/29), we were unable to measure all electromechanical durations:

-In two cats with cardiomyopathy, synchronization of the ECG and PCG device failed; so, measurements of EMAT, EMW and QS2 were not possible.-In one cat with cardiomyopathy, CHF and atrial fibrillation, electromechanical activity could not be assessed because S1 and S2 heart sounds could not be determined.-In one healthy cat, the ECG could not be used for further analysis due to unclear borders of low-voltage complexes, and therefore QT, EMAT, QS2 and EMW could not be measured.

The following measured intervals did not differ significantly between cats with cardiomyopathy and healthy cats: RR, QRS, QT, EMAT, QS2, EMW and S1S2 (Table 2). 

In cats with cardiomyopathy, EMAT/RR, QS2/RR and S1S2/RR were significantly longer, and predicted QT significantly shorter, than in healthy cats (Table 2).

## 4. Discussion

This is the first study investigating the electromechanical activity of the feline heart while synchronizing the PCG pilot device with the ECG, which has proven to be a valuable tool for evaluating phonoelectrocardiographic intervals. In cats with cardiomyopathy, some phonoelectrocardiographic intervals normalized to the cardiac cycle differed significantly from those of healthy cats, whereas absolute phonoelectrocardiographic values did not differ between groups. Intervals normalized to heart rate by dividing them by the RR interval, such as EMAT/RR, QS2/RR and S1S2/RR, were significantly longer in cats with cardiomyopathy than in healthy cats. We hypothesize that the reason for the longer EMAT/RR is that hypertrophied myocardium requires more time to generate sufficient pressure to close the mitral valve. In human patients, a longer EMAT/RR is associated with CHF [2,3,4,34,35]. Moreover, an increase in EMAT/RR > 15% has been shown to correlate with lower LV ejection fraction, higher end-systolic and end-diastolic volume indices and worse outcomes [3,5]. In a prospective study of human patients with CHF, the EMAT-guided group with a treatment goal of a 15% reduction in EMAT/RR Kaplan–Meier curves showed significant differences between the EMAT-guided group and the standard symptom-guided group in terms of rehospitalization for CHF and all-cause mortality [4]. Furthermore, in dogs, phonoelectrocardiographic intervals measured with the “EkoDuo Digital Stethoscope with ECG” device changed with the progression of myxomatous mitral valve disease; however, differentiation of stages based on recordings was not possible [10].

In our study, there were no differences in QT and QTc between cats with cardiomyopathy and healthy cats. These findings are contrary to the results of a previous study in which the authors reported that prolongation of QT and QTc, i.e., log600 × QT/logRR, may indicate LV hypertrophy, and a longer QT interval was a negative prognostic factor [14]. In our study, QTc1 = log600 × QT/logRR [14,22] did not differ between cats with cardiomyopathy and healthy cats. In the study with healthy beagles [22], a formula with log600 was used because 600 was the average RR interval of beagle dogs. Therefore, we also compared QTc2 with log300 because 300 was the average RR of cats in our study. Additionally, the values of QTc corrected according to the formulas of Bazett [19] and Fridericia [20], i.e., QT/square root of RR and QT/cubic root of RR, respectively, did not differ between healthy cats and cats with cardiomyopathy in our study. On the other hand, QT predicted from heart rate [23] was longer in healthy cats than in cats with cardiomyopathy in our study, where the difference of the RR interval between healthy cats and cats with cardiomyopathy was not significant (*p* = 0.055). 

In our study, QS2 and S1S2 did not differ between cats with cardiomyopathy and healthy cats; however, QS2/RR was significantly longer in cats with cardiomyopathy in comparison to healthy cats. The interval QS2 represents the time between the start of electrical systole and the end of mechanical systole. Similarly, S1S2/RR (mechanical systole) was longer in duration in cats with cardiomyopathy than in healthy cats. These findings suggest that electromechanical and mechanical events take longer in the ventricle of cats with cardiomyopathy.

In this study, we used an acoustic stethoscope with an inserted microphone and high-resolution digital recording on the smartphone, and a digital wireless ECG device, rather than a commercial PCG + ECG device. The reason for this choice was that we did not want to be limited by the frequency bandwidth of commercial devices. The intervals between electrical and mechanical events can be also measured with the use of echocardiography. The echocardiographic tracing has a resolution of 10 ms at 100 frames/second, which means that the electromechanical intervals could be determined with an accuracy of only 10 ms. However, this level of precision is not sufficient, as the variabilities over a respiratory cycle are much smaller, as we have shown in the pilot studies [24,25,36,37]. To achieve greater temporal accuracy in the measurement of electromechanical intervals, we employed simultaneous digital recording of the electrocardiogram (ECG) and phonocardiogram (PCG). This approach allowed us to accurately track changes in these intervals over time. While some electronic stethoscopes provide ECG capabilities, it is important to investigate the synchronization of recordings in electronic stethoscopes with acoustic monitors, especially when used to measure durations of electromechanical events.

The regulation of the electrical and mechanical processes in the heart takes place through the autonomic nervous system in response to the body’s demands. Thanks to smart devices and precise multifunctional sensors, we now have the ability to gain detailed insights into the electromechanical interactions of the heart. Auscultatory findings can be optimized with modern medical technology by characterizing sounds through recording, visualization and automated analysis systems [38,39]. The digital era has brought forth numerous exciting new methods of PCG synchronized with the ECG, which can also serve as valuable teaching tools for medical and veterinary students [36,40]. Similarly, the educational function of simple, low-cost PCG systems consisting of a stethoscope, a microphone and a smartphone has been presented [41]. Furthermore, heart sounds and murmurs recorded with electronic stethoscopes can be emailed for telemedicine evaluation [42].

This study has several limitations. The major limitation is the small number of patients, which precluded comparison of cats with cardiomyopathy with and without CHF, and correlation between electromechanical durations and disease status (healthy cats and cats with cardiomyopathy). In addition, we were not able to measure all electromechanical durations in 24.1% of the cats. In two cats with cardiomyopathy, synchronization of the ECG and PCG device failed, so measurements of EMAT, EMW and QS2 were not possible. In these two recordings, we noticed a large discrepancy in the length of the ECG and PCG, significantly deviating from the expected 300 ppm (0.03%), which was only detected when the recordings were analyzed. Through testing, we found that this discrepancy occurs when the computer (ECG recording) or cell phone (PCG recording) screen goes to sleep, causing the device failing to record all data. This causes the interruption of the recording, and synchronization is no longer possible. This can be prevented by setting the computer or phone in the awake mode for the time of the recording. Methods such as touching the screen to “wake up” the application are not recommended. The measurement can also be disturbed by applications that are unexpectedly activated on the computer or phone, necessitating that they be disabled. A phone call or received message can also interrupt a PCG audio recording. Therefore, the devices must be dedicated exclusively to this task during recording, with Internet access disabled.

Nowadays, the advancements in smartphones, applications, databases and algorithms for heart sound analysis have enabled not only the simple applications of phonocardiography [43,44,45], but also simultaneous ECG recording, which allows accurate measurements of electromechanical events [24]. Modern technology, which includes wireless PCG [44,45,46], developed acoustic sensors [43] and new methods of simultaneous PCG and ECG [47], will certainly contribute to an even more remarkable assessment of cardiac dynamics in the future [26].

## 5. Conclusions

The PCG synchronized to the ECG pilot device proved to be a valuable tool for evaluating the electromechanical activity of the feline heart. In cats with cardiomyopathy, several intervals, including EMAT/RR, QS2/RR and S1S2/RR, were found to be significantly longer than in healthy cats. These findings suggest that electromechanical and mechanical events take longer in the ventricles of cats with cardiomyopathy.

## Figures and Tables

**Figure 1 sensors-23-08336-f001:**
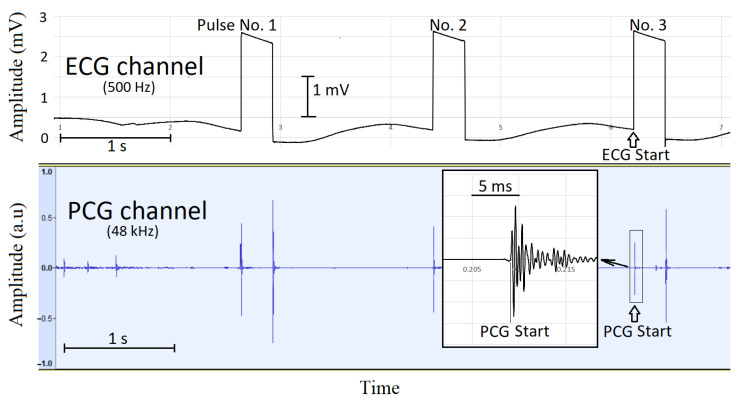
Three manually inserted synchronization pulses at the beginning of the recording. An equivalent set of three pulses is also inserted at the end of the recordings. The synchronization time instant for ECG and PCG recordings is taken at the beginning of pulse No. 3 at the beginning and end of the recordings. Each synchronization pulse is generated by a key-press and key-release of the switch. In the PCG channel, the key-press produces an oscillation waveform, as shown in the inserted picture, where the start of the oscillation is used as the synchronization time instant.

**Figure 2 sensors-23-08336-f002:**
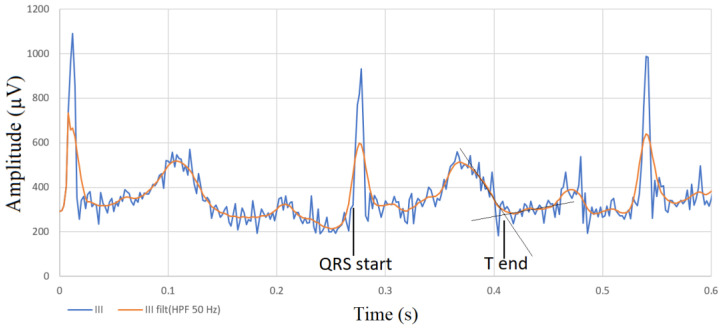
Unfiltered and filtered versions of the lead III in the ECG. The filtered version of the lead was used to define the end of the T wave (T end) as the intersection of a tangent to the descending part of the T wave with a line representing the local baseline. For the beginning of the QRS, the unfiltered version of the lead was used.

**Figure 3 sensors-23-08336-f003:**
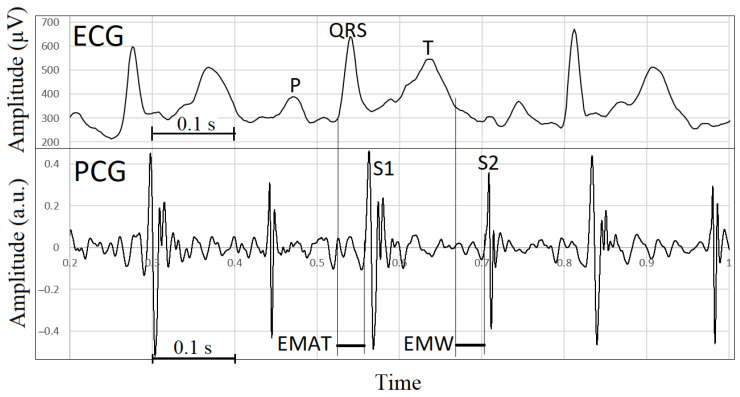
Matched ECG and PCG recordings. The onset of mechanical systole (onset of tone S1) always starts later than the onset of electrical systole (onset of QRS) by the interval of electromechanical activation time (EMAT). Also, the beginning of tone S2 (end of mechanical systole) in this case is behind the end of electrical systole (end of T wave). Therefore, the electromechanical window (EMW) has a positive value.

**Figure 4 sensors-23-08336-f004:**
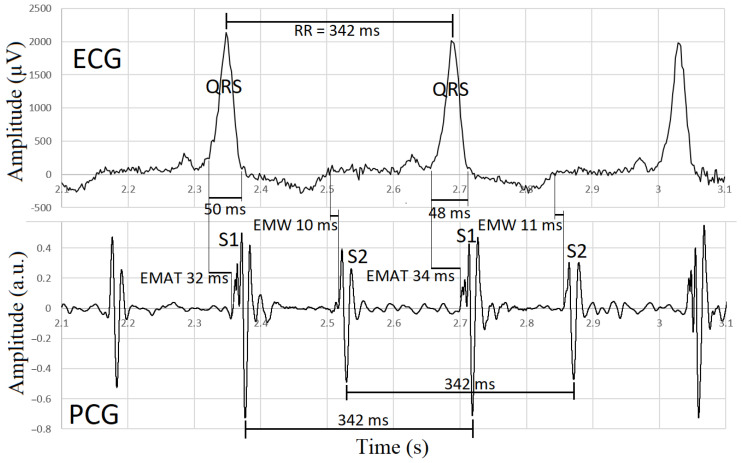
Electromechanical intervals electromechanical activation time (EMAT) and electromechanical window (EMW) for two consecutive heartbeats in a cat with cardiomyopathy and long QRS (here 50 and 48 ms). Note that the intervals measured in the ECG between the two heartbeats (RR interval) and between S1 and S2 beats have the same value (342 ms). The variability of the EMAT and EMW intervals is not greater than the temporal resolution of the ECG recording (2 ms).

**Table 1 sensors-23-08336-t001:** Basic characteristics of healthy cats and cats with cardiomyopathy.

	Healthy Cats	Cats with Cardiomyopathy
Number	12	17
Age (years; median, IQR)	4.18 (2.55–5.78)	7.80 (3.40–11.15)
Weight (kg; mean ± SD)	4.74 ± 1.24	5.01 ± 1.84
Female	9	6
Male	3	11
Breed	Domestic short-haired cat (*n* = 6), Siberian (*n* = 2), Maine Coon (*n* = 2), Bengal (*n* = 1), Sphynx (*n* = 1)	Domestic short-haired cat (*n* = 8), Sphynx (*n* = 6), Maine Coon (*n* = 1), British Shorthair (*n* = 2)

**Table 2 sensors-23-08336-t002:** Phonoelectrocardiographic data of healthy cats and cats with cardiomyopathy.

Variable	Healthy Cats (*n* = 12)	Cats with Cardiomyopathy (*n* = 17)	*p*
Heart rate (bpm) *	192.9 ± 23.4	211.8 ± 25.4	0.052
RR (ms) *	315.5 ± 40.6	287.2 ± 35.1	0.055
QRS (ms) **	24.0 (20.0–27.5)	28.0 (23.0–30.0)	0.070
QT (ms) *	161.6 ± 11.5	157.6 ± 13.6	0.462
QTc1 (ms) *	179.2 ± 10.6	178.2 ± 13.2	0.856
QTc2 (ms) *	159.7 ± 9.4	158.9 ± 11.8	0.856
QTc3 (ms) *	236.1 ± 12.6	239.1 ± 16.4	0.636
QTc4 (ms) *	285.6 ± 15.6	294.6 ± 20.3	0.250
QTpredicted (ms) *	187.8 ± 14.9	175.6 ± 13.4	**0.048**
QTdiff (ms) *	−26.2 ± 1.3	−18.0 ± 11.7	0.098
EMAT (ms) *	21.9 ± 9.3	28.4 ± 8.6	0.083
EMAT/RR *	0.069 ± 0.026	0.098 ± 0.028	**0.017**
QS2 (ms) *	171.1 ± 20.9	177.2 ± 19.2	0.467
QS2/RR *	0.542 ± 0.038	0.605 ± 0.048	**0.002**
EMW (ms) *	11.5 ± 17.6	17.6 ± 15.8	0.403
EMW/RR *	0.033 ± 0.057	0.058 ± 0.057	0.305
S1S2 (ms) *	149.2 ± 17.4	148.1 ± 15.7	0.873
S1S2/RR *	0.475 ± 0.034	0.514 ± 0.044	**0.017**
QT/S1S2 *	1.081 ± 0.130	1.082 ± 0.108	0.971

EMW: electromechanical window, interval measured from the end of the T wave to the start of the S2 sound (= (QS2) − QT); QRS: QRS or ventricular depolarization, from the onset of the Q wave to the end of the S wave; QS1 (EMAT): interval measured from the start of the Q wave to the start of the S1 sound (electromechanical activation time); QS2: interval measured from the start of the Q wave to the start of the S2 sound; QT: interval measured from the start of the Q wave to the end of the T wave (electrical systole); QTc1 = log600 × QT/logRR.; QTc2 = log300 × QT/logRR; QTc3= QT/cube root of RR; QTc4 = QT/square root of R; QTpredicted = 0.41845798 − (0.00181963 × HR) + (0.00000313 × HR^2^); QTdiff = QT − QTpredicted; RR: interval measured between the peaks of the R waves; S1S2: interval measured between S1 and S2 (mechanical systole); * normally distributed data expressed as mean ± standard deviation; ** non-normally distributed data expressed as median (interquartile range: 25th to 75th percentile); *p*-values less than 0.05 were considered significant (bold).

## Data Availability

All data are available from the corresponding author on reasonable request.

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
