# Peer review of "Cardiac Electromechanical Activity in Healthy Cats and Cats with Cardiomyopathies"

_sensors, 2023, doi:10.3390/s23198336_

Round 1
Reviewer 1 Report
Dear authors,
I will start by congratulating you for this great and inspiring manuscript.
My comments are minor:
line 92: please explain why a 12-lead ECG is needed for this study
line 135-136: No need to write "QS1 is the electromechanical activation time, and EMW is the electromechanical window" because you already described the abbreviation in the introduction section.
line 171: Throughout the results section you mention "lead III"(see line 215 and 226), and not "lead II" was used for the ECG measurements. How do you explain it?
Kind regards,
Reviewer 2 Report
The paper by Brložnik et al. describes the measurements of electromechanical parameters of the heart based on synchronous recording of electrocardiographic and phonocardiographic signals. The authors presented data on these measurements in healthy cats and cats with abnormal cardiac structure and found the differences in some of the temporal characteristics expressed as fractions of cardiac cycle (divided by electrocardiographic RR interval). I have some comments concerning general design and methodology.
General comments
1. The aim of the study is presented as: "We aimed to evaluate cardiac electromechanical activity in healthy cats and cats with cardiomyopathies using a digital 12-lead high-resolution ECG device with wireless data transmission synchronized with a non-commercial PCG pilot device. We looked for possible differences in electromechanical activity between cats with cardiomyopathy and healthy cats."
This formulation is not quite clear. What kind of useful information is to be derived from this comparison? Furthermore, the stated aims would shape methods and design. Is it to be used for diagnostics? If yes, the measured characteristics should be analyzed as cardiomyopathy predictors. Should this approach outperform or replace the existing methods, is it just to test an instrument? If yes, the data should be compared with those obtained by conventional methods. Is it to study how cardiomyopathy affects physilogical properties that are assessed by ECG-PCG (e.g. EMAT)? If yes, more meticulous measurements and analysis should be performed. See comments on methodological issues.
Moreover, if the authors have in mind the above or other aims, they should provide a background for these aims.
2. Diagnosis of cardiomyopathy is questionable. In the present article, it is stated as: "cardiomyopathy was diagnosed if the LV wall was thicker than 6 mm [29–31]. Hyperthyroid and systemically hypertensive cats with mild LV hypertrophy were excluded from the study".
Cardiomyopathy should not be considered as just abnormal heart size. It is defined as a myocardial disorder in which the heart muscle is structurally and functionally abnormal, in the absence of coronary artery disease, hypertension, valvular disease and congenital heart disease sufficient to cause the observed myocardial abnormality. Were these potential causes of myocardial abnormalities excluded? It looks like that only hypertensive animals with mild hypertrophy were excluded. What about hypertensive cats with severe hypertrophy? What about valvular disorders? Coronary artery diseases? I am not sure about hyperthyroidism as well, is it necessary to exclude this pathology?
3. Methodology. Synchronization (see also Fig. 1) concerns me. I would expect that oscillations in PCG elicited by external pulses used for synchronization should be delayed in respect to the electrical signal corresponding to these pulses recorded by ECG. This my concern is supported be the observation of the EMAT as long as QRS duration (Table 2). I would expect EMAT to be approximately twice shorter.
4. Figure 2 clearly demonstrates that ECG signal is very noisy. Exact determination of the onset of QRS complex (critically necessary for the measurement of EMAT) by this method is unreliable.
5. Normalization of the temporal variables by RR interval brings forth questions.
Cycle length correction for the QRS-related measures is unreliable especially taking into account the noisy ECG signal (see the comment above). Consider that the study (ref 15) reports the QRS duration change of only 1.25 ms for 100 ms change in RR!
Then, the authors did several QT corrections. What is the reason for this abundance? Strictly speaking, if the authors suspect that QT interval should be corrected by several different calculations, one may expect that all other rate corrected temporal variables are to be corrected by the same several calculations.
Moreover, the authors reported significant differences between the groups in QT/RR. Actually, it appears to be one more QT correction method, but this approach is not justified, whereas the justified methods yielded no significant differences.
Collectively, it looks like that the differences in RR-corrected temporal variables very probably stem from the fact that the sick animals had nearly significantly shorter RR intervals (higher heart rate).
Specific minor comments
6. Lines 144-145: "the beginning of the Q wave, the beginning of the QRS complex..." Why not simply the "beininning of the QRS"?
7. Lines 149-150: "Tachycardia was diagnosed if the heart rate exceeded 240 beats per minute" What for? To exclude animals with tachycardia? Other reasons?
8. Lines 319-321: "The reason for this choice was to avoid being limited by the characteristics of a commercial device". Which characteristics?
9. Figures. Axes designations are missing.
10. Table 2. Some data look like to have nonparametric distribution (e.g. EMW 11.5 +/- 17.6) and should be presented appropriately.
Reviewer 3 Report
In this study the authors investigate the heart electromechanical activity in healthy cats and cats with cardiomyopathy with phonocardiography (PCG) synchronized to an electrocardiography (ECG). The results showed that the methodology seems to be a valuable tool for evaluating the electromechanical activity of the feline heart and extends the frontiers we have now in both veterinary cardiology in particular and feline medicine in general.
The study is robust and well desgined. The results are properly explained and their insertion into the literature and their potential impact are discussed.
I consider the paper might be found interesting for the journal readers and I recomend it for publication.
Round 2
Reviewer 2 Report
I believe that the authors did their best to improve their article. That is why I do not insist on further revisions, although some concerns of mine persist. I also acknowledge the efforts of the authors and their meticulous work. The attached photograph with some comments would be very instructive as a supplementary figure.